# Cross Sections and Rate Coefficients for Rovibrational Excitation of HeH$^+$ Isotopologues by Electron Impact

**Mehdi Ayouz [1,*,†] and Viatcheslav Kokoouline [2,†]**

[1]    LGPM, CentraleSupélec, Université Paris-Saclay, 8-10 rue Joliot-Curie, 91 190 Gif-sur-Yvette, France
[2]    Department of Physics, University of Central Florida, Orlando, FL 32816, USA
*    Correspondence: mehdi.ayouz@centralesupelec.fr; Tel.: +33-175-316-603
†    These authors contributed equally to this work.

**Abstract:** Cross sections and thermal rate coefficients for rotational and vibration excitation of the four stable isotopologues of the $^4$HeH$^+$ ion by electron impact are presented. The data are calculated using a previously developed theoretical approach. The obtained rate coefficients are fitted to analytical formulas with the 10–10,000 K interval of applicability. These present results could be useful in tokamak plasma and astrophysical modeling and can help in the detection of these species in the interstellar medium.

**Keywords:** helium hydride ion; isotopologues; rovibrational excitation; R-matrix; quantum-defect theory; interstellar medium

## 1. Introduction

Over the past decades, it was suggested that the hydrohelium (helium hydride) cation HeH$^+$ can be present and observed in a number of astronomical environments, and particularly, in the planetary nebulae NGC 7027 [1–3]. In the interstellar medium (ISM), it is the process of radiative association of He and H$^+$ or of He$^+$ and H that forms the ion [4,5]. However, only very recently, the presence of the ion was confirmed by Güsten et al. [6] with the observation of the rotational ground-state transition of HeH$^+$ in the planetary nebula NGC 7027. These observations were made possible due to advances in terahertz spectroscopy [7] and high-altitude observatories [8]. Previously, a transition with a wavelength very similar to that of the $j = 1 \rightarrow 0$ rotational transition was found by Liu et al. [9]. However, further analysis suggested that it was actually caused by the CH molecule [10].

HeH$^+$ is easily formed in helium–hydrogen plasma, and in particular, in the hydrogen fusion reaction. The ion, with its isotopologues, plays an important role in the chemistry taking place in tokamaks, especially in the divertor region of the devices. Rovibrationally excited states, formed in collisions of HeH$^+$ with electrons, can be used for plasma diagnostics. The other process involving the HeH$^+$ isotopologues and electrons and taking place in the divertor and near walls of the reactors is the process of dissociative recombination. The process removes the ions from the plasma creating neutral atoms, which contribute to the damage of reactor walls.

There have been several experimental and theoretical studies [11–16] reporting cross sections for the dissociative recombination. Cross sections for vibrational excitation and de-excitation of the three lowest vibrational states of the ion by electron impact were also calculated previously [17]. However, the presence of vibrational resonances in the collisional spectra was ignored in that study. Čurík and Greene [18] have recently reported cross sections for rotational excitation of HeH$^+$ collisions, where the Rydberg series of rovibrational resonances were accounted for.

Data relevant to the other isotopologues $^4$HeD$^+$, $^3$HeH$^+$, and $^3$HeD$^+$ could also be useful for plasma modeling and diagnostics in fusion reactors. In this respect, cross sections as well as rate

coefficients for collisions of the $HeH^+$ isotopologues with electrons are needed. To our knowledge, there is no such theoretical or experimental data available for the $HeH^+$ isotopologues.

In a previous study [19] (hereafter referred to as paper I), we reported cross sections and rate coefficients for vibrational excitation for transitions between the five lowest vibrational levels. In a further study [20] (hereafter referred to as paper II), we presented similar data for rotational transitions in collisions of $^4HeH^+$ with electrons. In the present study, as a follow-up of papers I and II, we determine cross sections and rate coefficients for vibrational and rotational (de-)excitation for collisions of the four stable $HeH^+$ isotopologues with electrons.

The rest of the article is organized in the following way. The next section briefly discusses the theoretical approach used in the present calculation. A detailed description of the approach is presented at length in papers I and II, so we restrict ourselves here only to underline its major ideas. In Sections 3 and 4, the obtained rate coefficients for vibrational and rotational (de-)excitation are discussed and compared with the data available in the literature. Section 5 concludes the study.

## 2. Theoretical Approach

Similarly to papers I and II, the present theoretical method uses the UK R-matrix code [21,22] with the Quantemol-N interface [23] and some elements of the quantum defect theory (MQDT) [24–26]. The same parameters (the basis and orbital spaces, the R-matrix size, etc.) as in Paper II were employed in the electron-scattering calculations. As a first step in the theoretical approach, the body-frame scattering matrix $\hat{S}^\Lambda(R)$ is obtained numerically for a number of internuclear distances $R$ from $R = 0.85$ to $R = 3.95$ with a step of 0.05 bohr. At the second step, vibrational wave functions $\psi_v(R)$ for the four isotopologues are computed by solving the Schrödinger equation for vibrational motion using a DVR-type method [27].

Energies for vibrational and rotational transitions for the four isotopologues are shown in Tables 1 and 2 and compared with available data [28,29]. Note that unlike the present study, where the `aug-cc-pVQZ` basis is employed, in paper I, we used the `cc-pVQZ` basis to compute the potential energy curve (see Figure 2 of paper I). As a result, the obtained vibrational energies for the $^4HeH^+$ ion are slightly different in the present study and paper I.

**Table 1.** Energies $\Delta_v = E_{v+1} - E_v$ for vibrational transitions $v \to v + 1$ and rotational constants $B_v$ of the $^4HeH^+$ and $^4HeD^+$ molecules used in this study and compared with data available in the literature. All values are in $cm^{-1}$.

| Level $v$ | $^4HeH^+$ | | | | $^4HeD^+$ | | |
|---|---|---|---|---|---|---|---|
| | $\Delta_v$ | $\Delta_v$ [29] | $B_v$ | $B_v$ [28] | $\Delta_v$ | $B_v$ | $B_v$ [28] |
| 0 | 2910.8 | 2911.0007 | 33.523 | 33.558 | 2309.9 | 20.326 | 20.349 |
| 1 | 2604.4 | 2604.1676 | 30.808 | 30.839 | 2125.7 | 19.061 | 19.084 |
| 2 | 2296.2 | 2295.5787 | 28.074 | 28.090 | 1941.5 | 17.795 | 17.814 |
| 3 | 1983.0 | 1982.0562 | 25.282 | 25.301 | 1755.9 | 16.515 | 16.532 |
| 4 | 1661.6 | 1660.3559 | 22.389 | 22.402 | 1567.8 | 15.212 | 15.226 |
| 5 | 1329.5 | 1327.7860 | 19.338 | 19.344 | 1376.1 | 13.872 | 13.884 |
| 6 | 986.6 | 984.3599 | 16.061 | 16.058 | 1179.7 | 12.480 | 12.490 |
| 7 | 641.6 | 639.1959 | 12.492 | 12.479 | 977.9 | 11.019 | 11.025 |
| 8 | 328.0 | 327.3615 | 8.638 | 8.621 | 771.2 | 9.469 | 9.471 |
| 9 | 115.2 | 116.1487 | 4.854 | | 562.8 | 7.810 | 7.808 |
| 10 | 25.6 | 24.4099 | 2.064 | | 362.4 | 6.041 | 6.036 |
| 11 | | | | | 192.0 | 4.216 | 4.217 |
| 12 | | | | | 79.6 | 2.541 | 2.557 |

**Table 2.** Energies $\Delta_v = E_{v+1} - E_v$ for vibrational transitions $v \to v+1$ and rotational constants $B_v$ of the $^3$HeH$^+$ and $^3$HeD$^+$ molecules used in this study. All values are in cm$^{-1}$.

| Level $v$ | $^3$HeH$^+$ | | $^3$HeD$^+$ | |
|---|---|---|---|---|
| | $\Delta_v$ | $B_v$ | $\Delta_v$ | $B_v$ |
| 0 | 2994.7 | 35.681 | 2422.7 | 22.514 |
| 1 | 2668.1 | 32.693 | 2218.4 | 21.036 |
| 2 | 2339.4 | 29.682 | 2013.8 | 19.555 |
| 3 | 2004.7 | 26.602 | 1807.5 | 18.056 |
| 4 | 1660.7 | 23.400 | 1598.0 | 16.525 |
| 5 | 1304.5 | 20.008 | 1383.8 | 14.944 |
| 6 | 937.4 | 16.345 | 1163.9 | 13.295 |
| 7 | 573.2 | 12.340 | 937.6 | 11.551 |
| 8 | 261.0 | 8.054 | 706.8 | 9.687 |
| 9 | 77.4 | 4.107 | 478.2 | 7.684 |
| 10 | 15.9 | 1.496 | 270.8 | 5.567 |
| 11 | 16.8 | 0.733 | 118.8 | 3.495 |
| 12 | 24.3 | 0.940 | 38.6 | 1.836 |

*Scattering Matrix for Rovibrational Excitations in the HeH$^+$ Molecule and Its Isotopologues*

The next step in the treatment is the vibrational and rotational frame transformations. If we neglect the rotational structure of the ion, which corresponds to an experiment where cross sections for vibrational transitions $v \to v'$ are averaged over all possible initial and summed over allowed final rotational states of the levels $v$ and $v'$, the theoretical cross section is obtained from the following scattering matrix:

$$\mathcal{S}^{\Lambda}_{\lambda'v'l',\lambda vl} = \left\langle \psi_{v'}(R) \left| S^{\Lambda}_{\lambda'l',\lambda l}(R) \right| \psi_v(R) \right\rangle, \tag{1}$$

where the brackets imply an integration over the vibrational coordinates. As a second step, the rotational frame transformation is accomplished using the matrix elements $\mathcal{S}^{\Lambda}_{\lambda'v'l',\lambda vl}$ of Equation (1), leading to the laboratory-frame scattering matrix

$$\mathcal{S}^{J}_{j'\mu'l'v',j\mu lv} = \sum_{\lambda\lambda'}(-1)^{l'+\lambda'+l+\lambda}C^{j'\mu'}_{l'-\lambda'J\Lambda'}C^{j\mu}_{l-\lambda J\Lambda}S^{\Lambda}_{l'\lambda'v',l\lambda v}, \tag{2}$$

where $J$ is the total angular momentum of the $e^-$-HeH$^+$ system, $j$, $\mu$ and $j'$, $\mu'$ are the angular momenta with their projections on the molecular axis of the target before and after the rotational excitation of HeH$^+$ (and its isotopologues), and $C^{j'\mu'}_{l'-\lambda'J\Lambda'}$ and $C^{j\mu}_{l-\lambda J\Lambda}$ are Clebsch–Gordan coefficients. A detailed derivation of Equation (2) is given in Appendix A of paper II.

The matrices of Equations (1) and (2) are energy-independent and do not describe vibrational and rovibrational Rydberg resonances present in the collisional spectra. The actual scattering matrices $\mathcal{S}^{phys}$ are obtained from those two matrices, applying the closed-channel elimination procedure [24,26] as discussed in paper I. The total energy $E$ of the system is the sum $E = E_{el} + E_{j\mu v}$ of the relative kinetic energy $E_{el}$ of a collision and the energy $E_{j\mu v}$ of the initial state of the target.

## 3. Rate Coefficients and Cross Sections for Vibrational (De-)Excitation

The cross section for purely the vibrational transition $v \to v'$ is [30]

$$\sigma_{v'\leftarrow v}(E_{el}) = \frac{\pi\hbar^2}{2m_e E_{el}} \sum_{\lambda'l'\lambda l} \left| \mathcal{S}^{phys}_{\lambda'l'v',\lambda lv} - \delta_{\lambda lv,\lambda'l'v'} \right|^2, \tag{3}$$

where $m_e$ is the reduced mass of the electron-ion system. Figure 1 demonstrates, as examples, the cross sections of Equation (3) for the $v = 3 \to v' = 0, 1, 2, 4$ transitions of $^4$HeH$^+$ (solid lines) and $^4$HeD$^+$ (dashed lines). At very low scattering energies, below 0.001 eV, the de-excitation cross sections behave

as $1/E_{el}$ according to the Wigner threshold law [31]. At higher energies, all the (de-)excitation cross sections vary significantly due to the presence of series of Rydberg resonances.

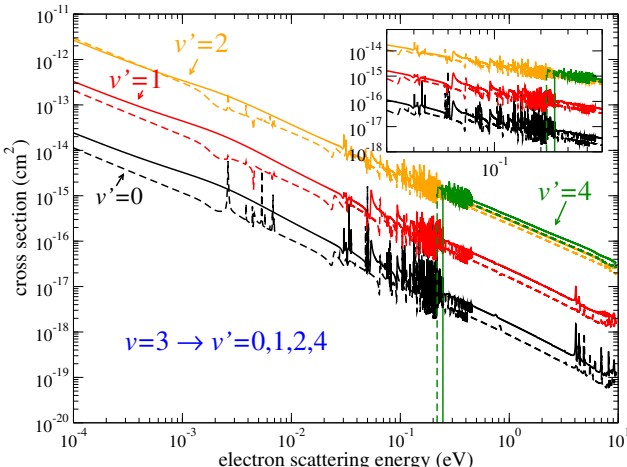

**Figure 1.** Cross sections of vibrational (de-)excitation from the vibrational level $v = 3$ to several other levels $v'$ of $^4$HeH$^+$ (solid lines) and $^4$HeD$^+$ (dashed lines).

Figures 2 and 3 show thermally averaged rate coefficients (see Equation (13) of paper I) computed for transitions between the lowest vibrational levels for the four $^4$HeH$^+$ isotopologues. The uncertainty of the rate coefficients for all transitions is about 5–30% for different temperatures. Due to the general $E_{el}^{-1}$ dependence of the cross sections, the calculated rate coefficients $\alpha_{v' \leftarrow v}$ behave as $1/\sqrt{T}$ as functions of temperature $T$ for de-excitation and as $\exp\left(-\Delta_{v'\,v}/T\right)/\sqrt{T}$ for excitation transitions, where $\Delta_{v'\,v} = E_{v'} - E_v$ is the excitation energy. Therefore, similarly to papers I and II, for convenience of use, the rate coefficients are fitted to the formula

$$\alpha_{v' \leftarrow v}^{fit}(T) = \frac{1}{\sqrt{T}} e^{-\frac{\Delta_{v'v}}{T}} P_{v'v}^{fit}(x),\tag{4}$$

where $P_{v'v}^{fit}(x) \approx P_{vv'}^{fit}(x)$ are functions weakly dependent on temperature interpolated by a cubic polynomial

$$P_{v'v}^{fit}(x) = a_0 + a_1 x + a_2 x^2 + a_3 x^3 \qquad \text{and} \qquad x = \ln(T),\tag{5}$$

with

$$\Delta_{v'v} = \begin{cases} E_{v'} - E_v > 0 & \text{for excitation}, \\ 0 & \text{for (de-)excitation}. \end{cases}\tag{6}$$

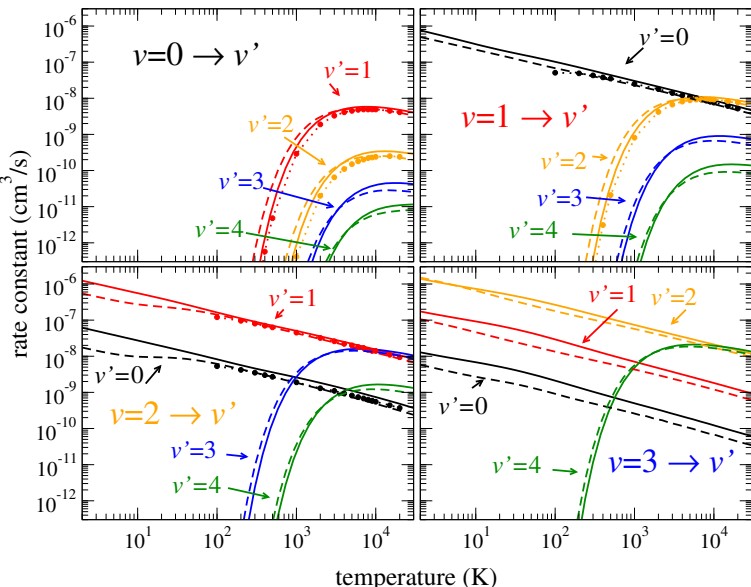

**Figure 2.** Examples of thermal rate coefficients for vibrational transitions in $^4$HeH$^+$ (solid lines) and $^4$HeD$^+$ (dashed lines). Results of a previous calculation [17] are shown by dotted lines with circles.

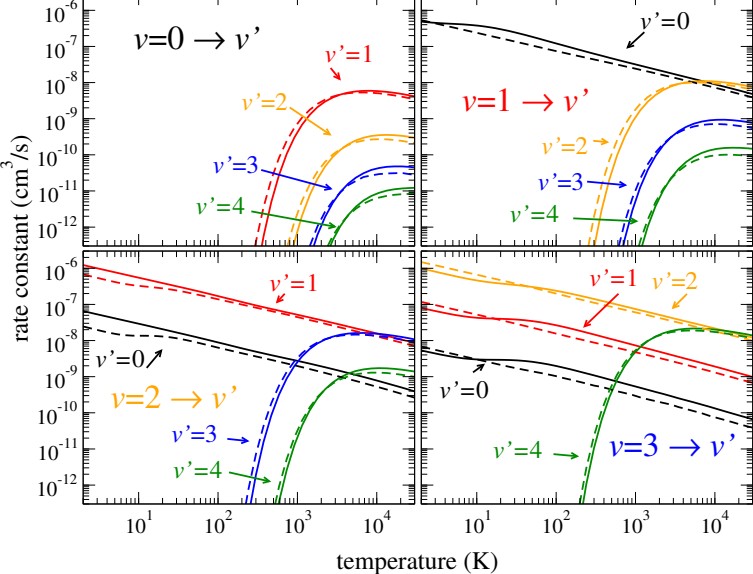

**Figure 3.** Same as Figure 2 for $^3$HeH$^+$ (solid lines) and $^3$HeD$^+$ (dashed lines).

The coefficients $a_i$ ($i = 0, 1, 2, 3$) are fitted for each pair of transitions $v' \leftrightarrow v$ and given in Tables 3–6. The numerical values of the coefficients $a_i$ in the Tables are such that they give the rate coefficients in units of cm$^3 \cdot$s$^{-1}$, with the temperature in fitting Equation (5) being in kelvin.

**Table 3.** Parameters $a_0$, $a_1$, $a_2$, and $a_3$ of the fitting polynomials $P_{vv'}^{fit}(x) = P_{v'v}^{fit}(x)$ of Equation (4) for $^4\text{HeH}^+$. The upper line in the header of the table specifies the pairs of initial and final vibrational levels for which the parameters are fitted. For convenience, we also specify (the second line of the header) the threshold energy $\Delta_{v'v}$ for the excitation process of the corresponding pair. For all excitation and de-excitation processes, the same parameters $a_i$ are used in Equations (4) and (5).

| $v$–$v'$ | 0–1 | 0–2 | 0–3 | 0–4 | 1–2 | 1–3 | 1–4 | 2–3 | 2–4 | 3–4 |
|---|---|---|---|---|---|---|---|---|---|---|
| $\Delta_{v'v}$ (K) | 4187 | 7935 | 11,238 | 14,091 | 3747 | 7050 | 9903 | 3303 | 6156 | 2853 |
| $a_0$ | $1.09 \times 10^{-6}$ | $8.84 \times 10^{-8}$ | $1.67 \times 10^{-8}$ | $3.59 \times 10^{-9}$ | $1.76 \times 10^{-6}$ | $2.26 \times 10^{-7}$ | $4.09 \times 10^{-8}$ | $1.81 \times 10^{-6}$ | $3.21 \times 10^{-7}$ | $2.54 \times 10^{-6}$ |
| $a_1$ | $-4.53 \times 10^{-8}$ | $-2.73 \times 10^{-9}$ | $3.98 \times 10^{-9}$ | $-1.08 \times 10^{-10}$ | $7.35 \times 10^{-11}$ | $5.34 \times 10^{-8}$ | $-1.39 \times 10^{-9}$ | $3.78 \times 10^{-7}$ | $-1.03 \times 10^{-8}$ | $-3.60 \times 10^{-8}$ |
| $a_2$ | $8.70 \times 10^{-9}$ | $6.49 \times 10^{-10}$ | $-8.73 \times 10^{-10}$ | $4.61 \times 10^{-11}$ | $-5.88 \times 10^{-9}$ | $-1.14 \times 10^{-8}$ | $5.87 \times 10^{-10}$ | $-5.85 \times 10^{-8}$ | $4.15 \times 10^{-9}$ | $1.62 \times 10^{-8}$ |
| $a_3$ | $-6.80 \times 10^{-10}$ | $-5.77 \times 10^{-11}$ | $4.17 \times 10^{-11}$ | $-3.88 \times 10^{-12}$ | $3.17 \times 10^{-10}$ | $5.42 \times 10^{-10}$ | $-5.13 \times 10^{-11}$ | $2.29 \times 10^{-9}$ | $-3.43 \times 10^{-10}$ | $-1.22 \times 10^{-9}$ |

**Table 4.** Same as Table 3 for $^4\text{HeD}^+$.

| $v$–$v'$ | 0–1 | 0–2 | 0–3 | 0–4 | 1–2 | 1–3 | 1–4 | 2–3 | 2–4 | 3–4 |
|---|---|---|---|---|---|---|---|---|---|---|
| $\Delta_{v'v}$ (K) | 3323 | 6381 | 9175 | 11,701 | 3058 | 5851 | 8378 | 2793 | 5319 | 2526 |
| $a_0$ | $7.12 \times 10^{-7}$ | $1.97 \times 10^{-8}$ | $7.91 \times 10^{-9}$ | $1.39 \times 10^{-9}$ | $6.90 \times 10^{-7}$ | $1.53 \times 10^{-7}$ | $1.28 \times 10^{-8}$ | $2.20 \times 10^{-6}$ | $1.41 \times 10^{-7}$ | $1.67 \times 10^{-6}$ |
| $a_1$ | $-2.48 \times 10^{-8}$ | $7.78 \times 10^{-9}$ | $6.27 \times 10^{-10}$ | $1.35 \times 10^{-10}$ | $1.28 \times 10^{-7}$ | $-7.04 \times 10^{-9}$ | $2.14 \times 10^{-9}$ | $-1.10 \times 10^{-7}$ | $1.20 \times 10^{-8}$ | $-4.68 \times 10^{-9}$ |
| $a_2$ | $7.50 \times 10^{-9}$ | $3.46 \times 10^{-10}$ | $-9.16 \times 10^{-11}$ | $1.35 \times 10^{-11}$ | $6.84 \times 10^{-9}$ | $1.29 \times 10^{-9}$ | $1.80 \times 10^{-10}$ | $9.19 \times 10^{-9}$ | $2.99 \times 10^{-9}$ | $3.31 \times 10^{-8}$ |
| $a_3$ | $-5.54 \times 10^{-10}$ | $-9.01 \times 10^{-11}$ | $1.13 \times 10^{-12}$ | $-2.15 \times 10^{-12}$ | $-1.48 \times 10^{-9}$ | $-1.00 \times 10^{-10}$ | $-3.35 \times 10^{-11}$ | $-3.32 \times 10^{-10}$ | $-3.69 \times 10^{-10}$ | $-2.86 \times 10^{-9}$ |

**Table 5.** Same as Table 3 for $^3\text{HeH}^+$.

| $v$–$v'$ | 0–1 | 0–2 | 0–3 | 0–4 | 1–2 | 1–3 | 1–4 | 2–3 | 2–4 | 3–4 |
|---|---|---|---|---|---|---|---|---|---|---|
| $\Delta_{v'v}$ (K) | 4308 | 8147 | 11,513 | 14,397 | 3838 | 7204 | 10,088 | 3365 | 6250 | 2884 |
| $a_0$ | $5.53 \times 10^{-7}$ | $9.34 \times 10^{-8}$ | $5.63 \times 10^{-9}$ | $3.88 \times 10^{-9}$ | $1.74 \times 10^{-6}$ | $8.98 \times 10^{-8}$ | $4.37 \times 10^{-8}$ | $1.29 \times 10^{-6}$ | $3.35 \times 10^{-7}$ | $2.60 \times 10^{-6}$ |
| $a_1$ | $4.11 \times 10^{-7}$ | $-1.81 \times 10^{-9}$ | $3.19 \times 10^{-9}$ | $-1.14 \times 10^{-10}$ | $2.06 \times 10^{-9}$ | $3.77 \times 10^{-8}$ | $-1.45 \times 10^{-9}$ | $1.31 \times 10^{-7}$ | $-1.03 \times 10^{-8}$ | $-3.54 \times 10^{-8}$ |
| $a_2$ | $-7.41 \times 10^{-8}$ | $4.50 \times 10^{-10}$ | $-8.26 \times 10^{-11}$ | $5.06 \times 10^{-11}$ | $-2.36 \times 10^{-9}$ | $-5.72 \times 10^{-10}$ | $6.33 \times 10^{-10}$ | $2.70 \times 10^{-8}$ | $4.24 \times 10^{-9}$ | $1.66 \times 10^{-8}$ |
| $a_3$ | $3.60 \times 10^{-9}$ | $-4.83 \times 10^{-11}$ | $-1.79 \times 10^{-11}$ | $-4.33 \times 10^{-12}$ | $5.86 \times 10^{-14}$ | $-2.41 \times 10^{-10}$ | $-5.56 \times 10^{-11}$ | $-3.23 \times 10^{-9}$ | $-3.52 \times 10^{-10}$ | $-1.25 \times 10^{-9}$ |

**Table 6.** Same as Table 3 for $^3\text{HeD}^+$.

| $v$–$v'$ | 0–1 | 0–2 | 0–3 | 0–4 | 1–2 | 1–3 | 1–4 | 2–3 | 2–4 | 3–4 |
|---|---|---|---|---|---|---|---|---|---|---|
| $\Delta_{v'v}$ (K) | 3485 | 6677 | 9574 | 12,175 | 3191 | 6089 | 8689 | 2897 | 5498 | 2600 |
| $a_0$ | $7.60 \times 10^{-7}$ | $3.03 \times 10^{-8}$ | $9.47 \times 10^{-9}$ | $2.01 \times 10^{-9}$ | $8.97 \times 10^{-7}$ | $1.67 \times 10^{-7}$ | $2.20 \times 10^{-8}$ | $2.10 \times 10^{-6}$ | $2.14 \times 10^{-7}$ | $2.12 \times 10^{-6}$ |
| $a_1$ | $-1.71 \times 10^{-8}$ | $8.90 \times 10^{-9}$ | $8.22 \times 10^{-11}$ | $-6.00 \times 10^{-11}$ | $9.90 \times 10^{-8}$ | $-1.77 \times 10^{-9}$ | $-1.07 \times 10^{-9}$ | $1.33 \times 10^{-8}$ | $-1.10 \times 10^{-8}$ | $-8.41 \times 10^{-8}$ |
| $a_2$ | $5.77 \times 10^{-9}$ | $-2.88 \times 10^{-10}$ | $3.09 \times 10^{-11}$ | $3.88 \times 10^{-11}$ | $4.12 \times 10^{-9}$ | $1.85 \times 10^{-10}$ | $6.28 \times 10^{-10}$ | $-8.84 \times 10^{-9}$ | $5.43 \times 10^{-9}$ | $3.13 \times 10^{-8}$ |
| $a_3$ | $-4.75 \times 10^{-10}$ | $-4.38 \times 10^{-11}$ | $-6.48 \times 10^{-12}$ | $-3.16 \times 10^{-12}$ | $-1.06 \times 10^{-9}$ | $-4.64 \times 10^{-11}$ | $-5.28 \times 10^{-11}$ | $3.98 \times 10^{-10}$ | $-4.37 \times 10^{-10}$ | $-2.22 \times 10^{-9}$ |

## 4. Rate Coefficients and Cross Sections for Rotational (De-)Excitation

The inelastic cross section for the rotational excitation or de-excitation process $j'\mu'v' \leftarrow j\mu v$ of a linear molecule by electron impact is obtained from the scattering matrix of Equation (2)

$$\sigma_{j'\mu'v'\leftarrow j\mu v}(E_{el}) = \frac{1}{2j+1}\frac{\pi}{k_j^2}\sum_{J,l,l'}(2J+1)\left|e^{i(l\pi/2+\sigma_l)}S_{j'\mu'l'v';j\mu lv}^{J,phys}e^{-i(l'\pi/2+\sigma_{l'})}\right|^2, \tag{7}$$

where $\sigma_l$ is the Coulomb phase shift. The derivation of the above formula is given in paper II.

In the ground electronic state of $^4\text{HeH}^+$, the projection $\mu$ of the electronic angular momentum on the molecular axis of the target is zero. Therefore, for scattering energies below the first excited electronic state $A^1\Sigma^+$ of $^4\text{HeH}^+$, $\mu = \mu' = 0$ in Equation (7). Figure 4 gives examples of the cross sections obtained with Equation (7) for the $j = 3 \rightarrow j' = 0, 1, 2, 4$ transitions of $^4\text{HeH}^+$ and $^3\text{HeH}^+$. The cross sections exhibit a strong resonant character for both molecular ions as well as for the two other isotopologues. These resonances are washed out when thermally-averaged rate coefficients are computed, leading to similar rate coefficients at high temperatures $T$, as shown by solid lines in Figures 5 and 6. Similar results are observed for $^4\text{HeD}^+$ and $^3\text{HeD}^+$, shown in the figures by dashed lines. However, the thermally averaged coefficients at low temperatures are sensitive to exact positions and widths of the lowest resonances because the integral over thermal velocities at low temperatures $T$ is determined only by small collision energies, $E_{el} \lesssim k_B T$. As a result, the rate coefficients for the $j = 2 \rightarrow j = 0$ transitions, for example, in Figures 5 and 6, are slightly different for different isotopologues.

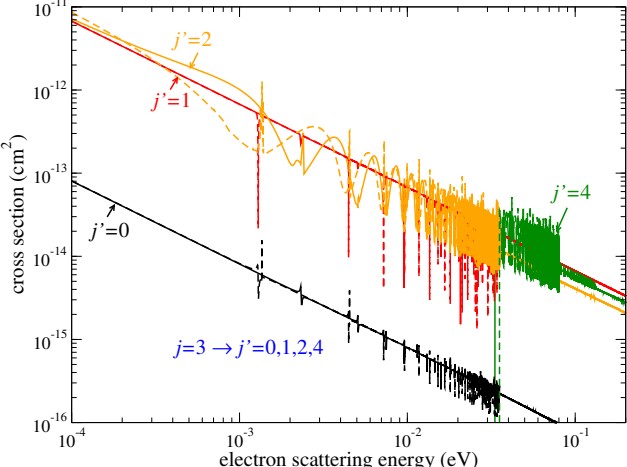

**Figure 4.** Cross sections of rotational (de-)excitation from the rotational level $j = 3$ to several other levels $j'$ of $^4\text{HeH}^+$ (solid lines) and $^3\text{HeH}^+$ (dashed lines).

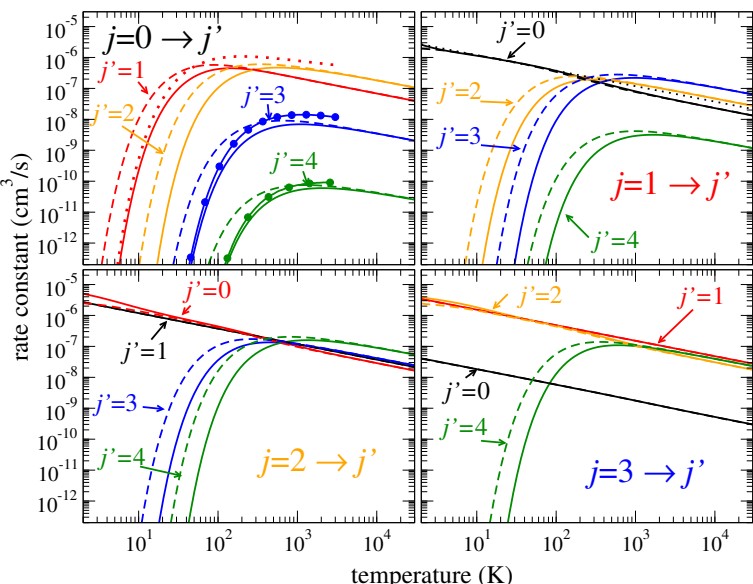

**Figure 5.** Thermally averaged rate coefficients for several rotational (de-)excitation transitions of $^4$HeH$^+$ (solid lines) and $^4$HeD$^+$ (dashed lines). Dotted lines in the upper-left panel are the calculations by Hamilton et al. [32], and lines with circles are those by Čurík and Greene [18] for $^4$HeH$^+$.

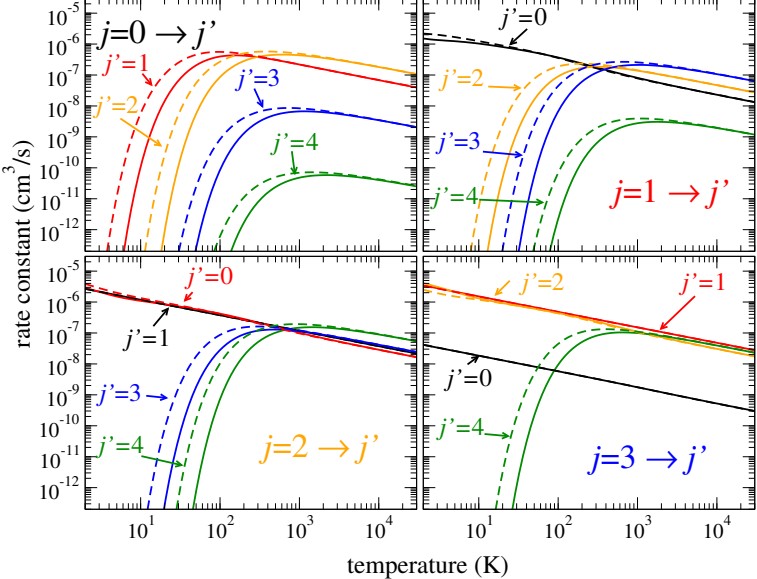

**Figure 6.** Same as Figure 5 for $^3$HeH$^+$ (solid lines) and $^3$HeD$^+$ (dashed lines). Rotational transition labels $j \to j'$ are shown in each panel.

In addition, the rotational rate coefficients behave approximately according to Equation (4), where $\Delta_{v'v}$ should be replaced with the rotational threshold energy, $\Delta_{j'j}$, and a quadratic polynomial is used in the fit. The probabilities for the direct $P^{fit}_{j'j}(x)$ ($j' \leftarrow j$) and the inverse $P^{fit}_{jj'}(x)$ ($j \leftarrow j'$) processes are related to each other by the relative degeneracy factor

$$P^{fit}_{j'j}(x) = \frac{2j'+1}{2j+1} P^{fit}_{jj'}(x). \tag{8}$$

The coefficients $a_i$ ($i = 0, 1, 2$) are fitted numerically for transitions $j' \leftrightarrow j$ and are given in Tables 7–10. Similarly to Tables in Section 3, the coefficients $a_i$ give the rate coefficients in Equation (5) in units of cm$^3$/s.

**Table 7.** Parameters $a_0$, $a_1$, and $a_2$ of the polynomial $P_{jj'}^{fit}(x)$ of Equations (4) and (5) for several pairs of initial and final rotational states for de-excitation $j \leftarrow j'$ of $^4$HeH$^+$, with $j < j'$. The probabilities $P_{j'j}^{fit}(x)$ for the opposite (excitation) process, $j \rightarrow j'$, are obtained from $P_{jj'}^{fit}(x)$, multiplying them with the factor $(2j'+1)/(2j+1)$ (see Equation (8)). For convenience, we also specify, in the second line of the table, the threshold energy $\Delta_{j'j}$ in units of temperature (K) for the excitation process of the corresponding pair. For the de-excitation processes, $\Delta_{j'j} = 0$.

| $j \leftarrow j'$ | $0 \leftarrow 1$ | $0 \leftarrow 2$ | $0 \leftarrow 3$ | $0 \leftarrow 4$ | $1 \leftarrow 2$ | $1 \leftarrow 3$ | $1 \leftarrow 4$ | $2 \leftarrow 3$ | $2 \leftarrow 4$ | $3 \leftarrow 4$ |
|---|---|---|---|---|---|---|---|---|---|---|
| $\Delta_{j'j}$ (K) | 96 | 289 | 578 | 964 | 192 | 482 | 868 | 289 | 675 | 385 |
| $a_0$ | $3.90 \times 10^{-6}$ | $3.79 \times 10^{-6}$ | $5.80 \times 10^{-8}$ | $4.47 \times 10^{-10}$ | $7.36 \times 10^{-6}$ | $4.86 \times 10^{-6}$ | $7.60 \times 10^{-8}$ | $5.79 \times 10^{-6}$ | $5.36 \times 10^{-6}$ | $3.29 \times 10^{-6}$ |
| $a_1$ | $-1.11 \times 10^{-7}$ | $-1.58 \times 10^{-8}$ | $3.35 \times 10^{-10}$ | $-2.17 \times 10^{-12}$ | $-8.57 \times 10^{-7}$ | $-8.43 \times 10^{-9}$ | $3.56 \times 10^{-10}$ | $-2.23 \times 10^{-7}$ | $8.55 \times 10^{-10}$ | $5.44 \times 10^{-7}$ |
| $a_2$ | $-6.57 \times 10^{-9}$ | $1.03 \times 10^{-9}$ | $-9.66 \times 10^{-11}$ | $7.92 \times 10^{-13}$ | $3.99 \times 10^{-8}$ | $2.68 \times 10^{-10}$ | $-9.86 \times 10^{-11}$ | $-8.13 \times 10^{-9}$ | $-4.29 \times 10^{-10}$ | $-5.98 \times 10^{-8}$ |

**Table 8.** Same as Table 7 for $^4$HeD$^+$.

| $j \leftarrow j'$ | $0 \leftarrow 1$ | $0 \leftarrow 2$ | $0 \leftarrow 3$ | $0 \leftarrow 4$ | $1 \leftarrow 2$ | $1 \leftarrow 3$ | $1 \leftarrow 4$ | $2 \leftarrow 3$ | $2 \leftarrow 4$ | $3 \leftarrow 4$ |
|---|---|---|---|---|---|---|---|---|---|---|
| $\Delta_{j'j}$ (K) | 58 | 175 | 350 | 584 | 116 | 292 | 526 | 175 | 409 | 233 |
| $a_0$ | $2.79 \times 10^{-6}$ | $3.76 \times 10^{-6}$ | $5.55 \times 10^{-8}$ | $4.42 \times 10^{-10}$ | $3.43 \times 10^{-6}$ | $4.74 \times 10^{-6}$ | $7.73 \times 10^{-8}$ | $3.66 \times 10^{-6}$ | $5.39 \times 10^{-6}$ | $5.86 \times 10^{-6}$ |
| $a_1$ | $2.67 \times 10^{-7}$ | $-5.03 \times 10^{-9}$ | $1.34 \times 10^{-9}$ | $-4.11 \times 10^{-12}$ | $2.84 \times 10^{-7}$ | $2.65 \times 10^{-8}$ | $4.68 \times 10^{-10}$ | $2.75 \times 10^{-7}$ | $-1.25 \times 10^{-8}$ | $-3.14 \times 10^{-7}$ |
| $a_2$ | $-3.52 \times 10^{-8}$ | $2.30 \times 10^{-10}$ | $-1.82 \times 10^{-10}$ | $1.12 \times 10^{-12}$ | $-3.78 \times 10^{-8}$ | $-2.11 \times 10^{-9}$ | $-1.35 \times 10^{-10}$ | $-3.71 \times 10^{-8}$ | $6.95 \times 10^{-10}$ | $1.99 \times 10^{-9}$ |

**Table 9.** Same as Table 7 for $^3$HeH$^+$.

| $j \leftarrow j'$ | $0 \leftarrow 1$ | $0 \leftarrow 2$ | $0 \leftarrow 3$ | $0 \leftarrow 4$ | $1 \leftarrow 2$ | $1 \leftarrow 3$ | $1 \leftarrow 4$ | $2 \leftarrow 3$ | $2 \leftarrow 4$ | $3 \leftarrow 4$ |
|---|---|---|---|---|---|---|---|---|---|---|
| $\Delta_{j'j}$ (K) | 102 | 308 | 616 | 1026 | 205 | 513 | 924 | 308 | 718 | 410 |
| $a_0$ | $2.31 \times 10^{-6}$ | $3.77 \times 10^{-6}$ | $5.83 \times 10^{-8}$ | $4.47 \times 10^{-10}$ | $4.24 \times 10^{-6}$ | $4.86 \times 10^{-6}$ | $7.67 \times 10^{-8}$ | $5.87 \times 10^{-6}$ | $5.41 \times 10^{-6}$ | $6.84 \times 10^{-6}$ |
| $a_1$ | $4.23 \times 10^{-7}$ | $-1.03 \times 10^{-8}$ | $2.04 \times 10^{-10}$ | $-1.62 \times 10^{-12}$ | $-9.42 \times 10^{-8}$ | $-1.53 \times 10^{-8}$ | $8.33 \times 10^{-11}$ | $-5.04 \times 10^{-7}$ | $-1.61 \times 10^{-8}$ | $-7.11 \times 10^{-7}$ |
| $a_2$ | $-4.59 \times 10^{-8}$ | $6.82 \times 10^{-10}$ | $-8.59 \times 10^{-11}$ | $7.16 \times 10^{-13}$ | $-5.05 \times 10^{-9}$ | $9.74 \times 10^{-10}$ | $-7.62 \times 10^{-11}$ | $2.25 \times 10^{-8}$ | $8.80 \times 10^{-10}$ | $3.47 \times 10^{-8}$ |

**Table 10.** Same as Table 7 for $^3$HeD$^+$.

| $j \leftarrow j'$ | $0 \leftarrow 1$ | $0 \leftarrow 2$ | $0 \leftarrow 3$ | $0 \leftarrow 4$ | $1 \leftarrow 2$ | $1 \leftarrow 3$ | $1 \leftarrow 4$ | $2 \leftarrow 3$ | $2 \leftarrow 4$ | $3 \leftarrow 4$ |
|---|---|---|---|---|---|---|---|---|---|---|
| $\Delta_{j'j}$ (K) | 64 | 194 | 388 | 647 | 129 | 323 | 583 | 194 | 453 | 259 |
| $a_0$ | $2.98 \times 10^{-6}$ | $3.79 \times 10^{-6}$ | $5.86 \times 10^{-8}$ | $4.43 \times 10^{-10}$ | $5.98 \times 10^{-6}$ | $4.81 \times 10^{-6}$ | $7.67 \times 10^{-8}$ | $3.79 \times 10^{-6}$ | $5.37 \times 10^{-6}$ | $3.44 \times 10^{-6}$ |
| $a_1$ | $2.62 \times 10^{-7}$ | $-1.13 \times 10^{-8}$ | $2.83 \times 10^{-10}$ | $-3.64 \times 10^{-12}$ | $-6.35 \times 10^{-7}$ | $-7.76 \times 10^{-9}$ | $5.76 \times 10^{-10}$ | $1.04 \times 10^{-7}$ | $-3.24 \times 10^{-9}$ | $4.46 \times 10^{-7}$ |
| $a_2$ | $-3.71 \times 10^{-8}$ | $6.05 \times 10^{-10}$ | $-1.03 \times 10^{-10}$ | $1.05 \times 10^{-12}$ | $3.24 \times 10^{-8}$ | $8.42 \times 10^{-10}$ | $-1.38 \times 10^{-10}$ | $-1.95 \times 10^{-8}$ | $-3.95 \times 10^{-11}$ | $-5.17 \times 10^{-8}$ |

## 5. Conclusions

We presented cross sections and thermal rate coefficients for rotational and vibrational transitions in the stable isotopologues of the $HeH^+$ ion caused by electron impact. The differences observed in cross sections for the four isotopologues are due to different positions of vibrational and rotational levels of the target ion. The different positions of the levels produce Rydberg resonances in the collisional spectra that are situated at different energies. Different positions of individual resonances can significantly modify cross sections. This is especially important at low energies, as demonstrated in Figures 1 and 4. Very different cross sections at low collision energies lead to very different thermal rate coefficients at low temperatures.

Because the overall coupling between different vibrational and rotational channels is the same for all isotopologues, generally, widths of the resonances are comparable for the four isotopologues. This results in thermally averaged rate coefficients that are very similar in magnitude to each other for the four isotopologues. The only essential effect on the rate coefficients is due to a higher density of rovibrational levels and, as a result, a higher density of resonances in the collisional spectra for heavier isotopologues. This effect is evident in Figures 5 and 6 showing the coefficients for rotational excitation: For heavier isotopologues, the rotational excitation rate coefficients are, in general, higher. For vibrational transitions, the ratio of densities of vibrational resonances between different isotopologues is closer to unity compared to the rotational-level densities. Therefore, the isotope effect on the vibrational excitation coefficients is less important.

We extended our previous studies on $^4HeH^+$ to its isotopologues $^4HeD^+$, $^3HeH^+$, and $^3HeD^+$. The obtained results are important for hydrogen–helium plasma modeling and diagnostics and could contribute to the search of the $^4HeH^+$ isotopologues in astrophysical environments.

**Author Contributions:** All authors contributed equally to this work.

**Funding:** This research was funded by Grant No. PHY-1806915 of the National Science Foundation, the Thomas Jefferson Fund of the Office for Science and Technology of the Embassy of France in the United States and by the program "Accueil des chercheurs étrangers" of CentraleSupélec.

**Acknowledgments:** The authors are grateful to the referees for the constructive comments and improvement suggestions.

**Conflicts of Interest:** The authors declare no conflict of interest.

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
