# Peer review of "Cross Sections and Rate Coefficients for Rovibrational Excitation of HeH+ Isotopologues by Electron Impact"

_atoms, doi:10.3390/atoms7030067_

Round 1

Reviewer 1 Report

Referee report: Cross sections and rate coefficients for rovibrational excitation of HeH+ isotoplogues by electron impact.

Submitted to Atoms (2019) by M. Ayouz and V. Kokoloouine

The article is a continuation of the theoretical studies on ro-vibrational excitation and de-excitation by electron impact with HeH+ earlier published by M Ayouz et al, Atoms, 4, 30 (2016) and M Khamesian et al Atoms, 6, 49 (2018). Here the cross sections and rate coefficients for different isotopologues of the HeH+ system are calculated.

The article is well written and the results could important for plasma modeling and diagnostics in fusion reactors. However, there are some concerns I would like the authors to address.

The HeH system has series of electronic resonant states that are Rydberg states converging to excited electronic cores. I understand that in the present study, these states are not considered. However, for collision energies reaching 10 eV, the electronic resonant states can be reached. This has to be clarified. Either the electron scattering energies should be smaller than the energy needed in order to capture into these resonant states, or the effects of the resonant states should be considered.

Page 2, 3rd paragraph: “in which the Rydberg series of vibrational resonances in HeH are accounted for” – specify what isotopologue of HeH was studied.

Page 3, 2nd paragraph: Below eq. (1) the electronic resonant states are discussed, however, it is not clear what the authors did.

Page 3, section 2.2: Why are the authors using a different method CCSD(T)/aug-cc-pVQZ  for computing the HeH+ potential energy curve than what they are using in the electron scattering calculations?

Page 4, below eq. (2): “vibrational coordinates” -> “internuclear distance”

Page 5, eq. (4) In eq. (3) the rotational frame transformation is performed and the scattering matrix has the labeling $S^J_{j’ \mu’ l’ v’,j \mu l v}$. Then the closed-channel elimination is performed and the physical scattering matrix is computed. Why has the physical scattering matrix in eq. (5) the labeling $S^{phys}_{\lambda’ l’ v’,\lambda l v}$?

Page 5, below eq. (4): add a reference to Wigner threshold law.

Page 5, fig 1. The inset in the figure is not improving the figure.

The main goal of the article is to compute the cross sections and rate coefficients for rovibrational (de-) excitation in electron scattering with different isotopologues of the HeH+ ion. However the isotope dependence is not commented or discussed. This is something I miss.

Author Response

We would like to thank the referee for the constructive comments and improvement suggestions. We have modified the manuscript according to referees' suggestions. We answer the referees' questions and describe changes made to the manuscript in the attached file.

Reviewer 2 Report

    This article is very important for the modeling of the Early Univers kinetics, and for that of several interstellar molecular clouds.

    I recommend publication, but also performing minor corrections, suggested in red at pages 1, 2 and 15.

Author Response

We would like to thank the referees for the constructive comments and improvement suggestions. We have modified the manuscript according to referees’ suggestions. We answer the referees' questions and describe changes made to the manuscript in the attached file.
